# Reactive Powder Concrete Mix Ratio and Steel Fiber Content Optimization under Different Curing Conditions

**DOI:** 10.3390/ma12213615

**Published:** 2019-11-04

**Authors:** Yunlong Zhang, Bin Wu, Jing Wang, Mo Liu, Xu Zhang

**Affiliations:** 1School of Transportation Science and Engineering, Jilin Jianzhu University, Changchun 130000, China; zhangyunlong@jlju.edu.cn (Y.Z.); wu616728257@163.com (B.W.); 2School of Civil Engineering, Jilin Jianzhu University, Changchun 130000, China; lium547573@163.com; 3Jilin JingCheng Engineering Detection Co., Ltd. Changchun 130033, China

**Keywords:** orthogonal test, compressive strength, splitting tensile strength, steel fiber-reinforced concrete

## Abstract

In this paper, a practical reactive powder concrete mixture ratio is created on the basis of an orthogonal experiment. Previous studies have combined the compressive and splitting tensile strengths of four categories of reactive powder concrete (RPC) for major materials. These categories include water/binder ratio, silica fume volume content, sand/binder ratio, and dosage of fly ash volume. The optimal mixing proportion of each factor was determined by analyzing the compressive strength of the RPC matrix. For this purpose, steel fiber was used as a reinforcing agent. The compressive and splitting tensile strength test results of steel fiber RPC were analyzed by comparing compound, standard, and natural curing. This was conducted to explore the improvement effect of different steel fiber contents on compressive performance, especially tensile strength of the RPC matrix. According to the results, the optimal steel fiber content was found to be 4% under the three curing conditions. The effect of compound curing on early strength was found to be greater in RPC than by natural or standard curing. However, the effect of late improvement is not obvious. Although standard curing is slightly stronger than natural curing, the performance under the latter can still meet engineering requirements.

## 1. Introduction

Reactive powder concrete (RPC) is a low energy consuming, eco-friendly, cement-based compound material with good fluidity, mechanics, and durability [1,2]. It is used as a gelling system to remove coarse aggregates using Portland cement and various industrial wastes. The resulting fine aggregates, fibers, and nano-mineral admixtures are then used as reinforcing phases with the help of conventional preparation techniques. The compound offers better compressive strength (especially tensile strength), durability, and impact toughness. Therefore, high-performance structures built using RPC material can greatly save the amount of material used, reduce the size of the structure, make the structure lighter, reduce energy consumption and pollution, and improve the durability and economy of the whole life of the structure. An RPC structure can greatly simplify or even completely eliminate ordinary steel reinforcement, streamline the construction process, be suitable for construction assembly, reduce cost, and possibly solve problems in existing bridges, such as fatigue cracking of bridge decks, fragile pavements, cracking and bending of traditional pre-stressed concrete beams, low assembling rate of small and medium-sized span fabricated bridges, etc. [3,4,5]. However, the current mix ratio of RPC is mostly formulated on the basis of scientific research and experimental experience. Even if the same mix ratio is adopted, RPC performance will be significantly diverse due to regional differences in materials and various curing conditions, which are non-conducive to the widespread use of commercial RPC. Therefore, the mix ratio of RPC from ordinary raw materials must be optimized.

Although there is no RPC standard maintenance system, researchers have reached a consensus on the main method of curing conditions, namely, thermal curing. The high curing temperature can effectively promote the hydration reaction inside the RPC matrix and accelerate mineral blending. In the volcanic ash reaction process, a dense reaction improves the matrix’s properties [6,7,8]. The pre-pressure on the specimen during curing also has a certain impact on RPC compressive strength. The compressive strength of the RPC matrix increased to 500 MPa by applying a pre-pressure of 60 MPa at a curing temperature of 250 °C [2]. Ipek [9] conducted a systematic study on the impact of preset pressure on the mechanical properties of RPC. The results indicated that compressive strength is high when pre-pressure is also high. The test results showed that micro-cracks in the RPC matrix will expand with increase in pre-pressure when the pre-pressure exceeds a certain range. This situation causes a decline in RPC compressive strength. The curing temperature and humidity of standard curing are maintained at a constant state, which is conducive to the formation of RPC body strength. However, the temperature of a natural curing environment cannot be kept stable, and air humidity is far less than that of standard curing. Compound curing is heat curing done over a period of time under the foundation of natural curing. Considering that current thermal curing is mostly limited to laboratory or prefabrication factories, RPC performance must be studied under conventional (natural and standard) and compound curing conditions to promote the connection between RPC and actual engineering. Thus, a reference for follow-up research and application of RPC is provided.

There is a very direct relationship between materials and fit ratio of RPC matrix strength. Compared with ordinary concrete, RPC, based on maximum compactness theory, has better aggregate gradation. Furthermore, the use of mineral admixture can not only effectively reduce the hydration heat of cement, but also effectively enhance the matrix strength of RPC on the premise of reducing the amount of cement. With the application of nanotechnology, many nanoscale materials have been proven to be helpful in strengthening the properties of RPC materials. There are currently several studies being conducted on nano silica [10,11], nano calcium carbonate [12], carbon nanotubes [13,14,15], metakaolin [16,17], rice husk ash [18], glass powder [19], and superfine cement and mineral powder. The results of studies on the mixture ratio are not the same, but most of them focus on one or more factors, such as water-binder ratio [20,21,22,23,24,25], sand-binder ratio [21,22,23,24], fiber content [21,22,23,24,25], fly ash content [22,23,25], and silica fume content [21,22,23,25]. With high compressive strength and tensile strength as the target, analyses of the mixing ratio in the current literature reveal that the effect of the proportion of steel fiber content and mineral admixture on compressive strength, especially tensile strength of the RPC matrix, is significant. In addition, the water/binder ratio and sand/binder ratio that affect the basic properties of concrete also account for a large proportion. When these nanoscale admixtures enhance the strength of the RPC matrix, the cost of RPC also increases, which is not conducive to its promotion and application in practical projects. How to use common raw materials to create an efficient RPC is the focus of this work; thus, determining the ratio of various materials is key. 

Fiber, which can improve mechanical properties and enhance the tensile and impact toughness of concrete, was added to RPC as a reinforcing agent. There are several studies on the kinds of fiber-reinforced RPC available worldwide: it is revealed that RPC performance is directly related to fiber content [26,27,28], different curing methods [29], elastic modulus [30], fracture strength, mechanical properties, and fiber surface characteristics [31,32]. As an early development step, adding steel fiber to concrete is an important means to improving concrete performance. That is, most researchers have used mechanical tests to study the compressive and flexural strengths of RPC and have focused on the basic mechanical properties of components and design calculation theory. At present, there are differences on the optimal content of steel fiber. Ji [29] states that the RPC crack resistance of the best steel fiber content is 3%. The study results reveal that steel fiber content has the most significant effect on the cracking resistant behavior of RPC. Fujikake [22] studied the strain of steel fiber RPC under tensile stress and obtained the strain rule of steel fiber RPC under tensile stress. Xue [31] reports that the optimal content of 13 mm steel fibers, obtained using a scanning electron microscope (SEM), is 1%. Jiang [32] combined the fluidity and mechanical properties of steel fiber RPC; their results indicate that the optimal content of 13 mm steel fiber is 2%–3%. Wille [33] studied the influence of different fiber content and shape on the tensile properties of RPC, and the results show that the tensile strength of RPC is proportional to the content of the same steel fiber shape. The test results indicate that the addition of steel fibers significantly improves load-carrying capacity, post-cracking stiffness, and cracking response, but decreases ductility. In addition, a study by Yoo [34] revealed that an increase in the length of smooth steel fibers and the use of twisted steel fibers led to improvements in post-peak response and ductility. In-depth studies on the tensile strength and microscopic mechanism of steel fiber-reinforced RPC [33,34,35,36] and different scales and kinds of steel fiber [32,37] are scarce, and existing literature on determining the RPC mixture has been mostly based on experience and numerous test results to determine whether the optimal dosage of steel fiber is relatively different. Even if the steel fiber specifications are the same, raw material preparation, mixed system, and maintenance conditions differ. Consequently, a significant difference is observed in optimal fiber content. Therefore, the compressive and splitting tensile strengths of RPC must be investigated with various mixing amounts and under different curing conditions for steel fibers that are widely used in practical projects. This step is necessary to determine the optimal steel fiber mixing amount and provide flexible and scientific data support for future engineering design and practical use.

In summary, there are many factors affecting the matrix strength of RPC materials. Therefore, this study used conventional materials and an orthogonal test (to study compressive strength consequence) to determine the optimal basic reference ratio of RPC under the following four factors: water/binder ratio, silica fume volume content, sand/binder ratio, and fly ash volume content. Steel fibers with a volume of 1%–5% were added to the RPC, and the five-volume steel fibers were subjected to basic curing tests for 7 and 28 days under natural, standard, and compound curing conditions. The compressive and splitting tensile strengths were studied, and the microscopic recorded data of RPC analyzed by using scanning electron microscopy.

## 2. Materials and Methods

### 2.1. Raw Materials

The test adopts P.II52.5 cement by Jilin Yatai Cement (Changchun, China). The cement inspection report (Table 1) complies with the General Portland Cement Testing Standard (GB175-2007) [38]. Silica fume produced by Dongyue Silicone Material (Zibo, China), fly ash produced by Datang Second Thermal Power (Changchun, China), and S95-grade mineral powder produced by Longze Water Purification Materials (Gongyi, China) were used. The test reports are presented in Table 2, Table 3 and Table 4. These materials were used in accordance with the technical specifications of the application of mineral admixtures (GB/T51003/2014) [39]. The superplasticizer used was HSC polycarboxylic acid high-performance water-reducing agent produced by Hongxia Polymer Materials (Qingdao, China). A fine aggregate with particle size below 0.18 mm was used; it was derived from natural river sand in Jilin Province—its fineness modulus was 2.158. The steel fiber was made of copper-plated steel fiber produced by Zhitai Steel Fiber Industry (Tangshan, China). The specifications and performance indexes are shown in Table 5. Figure 1 shows a picture of steel fiber.

### 2.2. Test Preparation and Curing

First, cement, silica fume, fly ash, mineral powder, and river sand were poured into the mixer and mixed dry for 2 min. Second, 80% of the mixture and water was slowly added to the mixer and stirred for 2 min. When the steel fiber was added, it scattered after 2 min of dry mixing. Third, the mixture of 80% superplasticizer and water was slowly added and stirred for 4 min. Finally, the remaining superplasticizer agent was added and stirred for 2 min.

This experiment adopted three curing methods, namely, indoor natural curing (temperature: 15 °C ± 2 °C), standard curing (temperature: 20 °C ± 2 °C, humidity: 95%), and compound curing. The processes are described in Table 6. 

### 2.3. Testing Program

Cube specimens of 100 mm × 100 mm × 100 mm were selected as compressive and splitting tensile strength specimens in accordance with GB/T31387-2015 [40]. Slump test was done using a standard slump test bucket and a test ruler. The loading rates of the compressive test and splitting tensile strength test were maintained between 1.2~1.4 MPa/s and between 0.08~0.1 MPa/s, correspondingly. SYE-3000B press (Hydraulic Press of New Testing Machine Co., Ltd., Changchun, China) was used for loading. A scanning electron microscope TM3030 of Hitachi was used to analyze the micromorphological structure of the material. 

### 2.4. Optimize RPC Mix Ratio

An orthogonal experimental efficiently and rapidly selects representative test points from comprehensive tests. The selected test points are “evenly dispersed and can be easily compared”. Therefore, when dealing with multi-factor and multi-level problems, as shown in Figure 2, the samples selected from the orthogonal experiment had uniform dispersion degree. According to the characteristics of this experiment, the number of experiments can be reduced to the greatest extent, and the results are conducive to the scientific judgment and analysis of the experimental results.

Taking the basic mixing ratios as the research objective, four factors, namely, water/binder ratio, silica fume volume content, sand/binder ratio, and fly ash volume content, were selected to find the optimal reference ratio of the abovementioned factors by adopting an orthogonal test with four factors and three levels. The level of each factor is shown in Table 7, and the proportion of the orthogonal experiment is shown in Table 8.

## 3. Mechanical Properties of RPC 

### 3.1. Mix Ratio Optimization Results

According to Figure 3, from the perspective of overall strength, compound curing was found to be greater than standard curing, which was, in turn, found to be greater than natural curing. In the early stage, the advantages of compound curing are obvious in comparison with the 7-day/28-day strength due to the participation of thermal curing. However, in preparation of curing for 28 days, the intensity of standard and natural curing gradually increase with curing time. The gap between its intensity and thermal curing decreases, and the difference between standard and natural curing is small.

### 3.2. Analysis of Range

The range analytical method in the orthogonal test was used to analyze the results in Table 9 and calculate the range influencing values of the water/binder ratio, sand/binder ratio, silica fume content, and fly ash content on 28-day compressive strength of active powder concrete. Table 10 shows the results of this analysis. The magnitude of the extreme value R indicates the influencing degree of this factor on compressive strength. The sand/binder ratio has the greatest influence, followed by silica fume, water/binder ratio, and fly ash. The data obtained from the range analysis to analyze the influencing law of water/binder ratio, sand/binder ratio, silica fume content, and fly ash content on RPC matrix strength specifically are illustrated in Figure 4, Figure 5, Figure 6 and Figure 7.

Figure 4 shows that different curing conditions significantly influence the compressive strength of fiber-free RPC matrix. RPC strength under the three curing conditions increases with decrease in the water/binder ratio. RPC strength significantly increases in the compound curing mode with low water/binder ratio and hot curing participation. Under the dense condition, the low water/binder ratio results in RPC deterioration and vulnerability. However, an excessive water/binder ratio leads to an increase in water content in the mixture. In addition to causing severe hydration reaction and increasing the hydration degree of cement, the rest of the water dilutes the mixture and exists in a free form. When excess water migrates into the matrix, it produces micropores that adversely affect the matrix strength. While designing the RPC mix ratio, the water/binder ratio must be reduced to ensure matrix strength, and a low water/binder ratio must be considered to diminish RPC workability.

Figure 5 demonstrates that, under the three curing conditions, the matrix strength initially increases and then decreases with a rise in silica fume content. When silica fume content accounts for 13% of the cementing material, the compressive strength of the RPC matrix is best improved. The compressive strength of the RPC matrix is best improved with compound curing. Silica fume is mainly used in pores of cement slurry and interspaces of cement particles. The surface-active substances of silica fume enable it to produce a “ball” effect among cement particles and improve the fluidity of the matrix mixture. Under the thermal curing condition, the volcanic ash effect can transform calcium hydroxide, which adversely affects strength, into C–S–H gel and use it for cement hydration products, effectively strengthening matrix strength. However, silica fume content cannot be blindly increased. Excess silica fume absorbs considerable water, given its large specific surface area, thereby resulting in an increase in water requirement. Excess water in confined pores of concrete after hydration will lead to a decrease in its relative density and affect its strength. This conclusion is similar to that of Liang [22] and Chen [24].

Figure 6 demonstrates that RPC strength under compound curing conditions gradually decreases, whereas its strength under standard and natural curing slowly increases with fly ash content. The addition of fly ash can improve the grain size distribution of the gelled material system and significantly enhance the slurry filling compactness. The particle size distribution of fly ash in concrete with silica fume can reach particles during pore filling and enter the discharge hole of free water. Accordingly, the concrete under the condition of low water/binder ratio shows good liquidity, increases the density of the gel system, and reduces the internal matrix porosity, cement dosage, and environmental pollution. In compound curing, the addition of fly ash with relatively weak activity will slightly affect the pozzolanic reaction in the matrix, resulting in the loss of some early strength.

Figure 7 shows that the variation rules of the RPC matrix strength and sand/binder ratio under the three curing conditions are similar. That is, the compressive strength of the matrix decreases with increase in the sand binder ratio. The influence of the sand/binder ratio on the compressive strength of active powder concrete is related to the mixture compactness. The sand/binder ratio is small, and the average slurry thickness increases. Consequently, the entire mixture is connected to an effective whole, the bonding force between sand and slurry is significantly increased, and the compressive strength obviously increases. However, this situation does not mean that the compressive strength will be high because the sand/binder ratio is low. The influence of sand skeleton on RPC strength remains dominant.

### 3.3. Analysis of Optimum RPC Mixture Ratio

The analytical results of the range test indicate that the influencing degree of the abovementioned four factors on RPC matrix strength is sand/binder ratio > silica fume > water/binder ratio > fly ash. Under the different curing conditions, a new mix ratio combination (Table 10) can be obtained by range analysis based on the test results. The mix ratio mechanical property test results are shown in Table 11.

Under standard or natural curing conditions, the optimal mix ratio obtained by range analysis is consistent with that of the fifth group in the orthogonal test. The experimental results are slightly different. However, under the hot water curing condition, the compressive strength of the matrix is highest when the water/binder ratio is 0.16. In contrast, appropriately reducing the volume content of fly ash is conducive to increasing the pozzolanic and hydration reaction, thus promoting the improvement of RPC matrix strength. The analysis of the two test results indicates that the optimal reference ratio of each factor under the natural and standard curing conditions are presented as follows: 0.18 water/binder ratio, 0.7 sand/binder ratio, 20% fly ash volume content, and 13% silica fume volume content. The optimal water/binder ratio of compound curing is 0.16, the sand/binder ratio is 0.7, the fly ash volume content is 10%, and silica fume volume content is 13%.

## 4. Design of Steel Fiber Content

The optimum proportion of each admixture was adopted on the basis of a previous study, wherein a 12-mm steel fiber was added as a reinforcing agent. Based on the different curing systems, given the fiber RPC compressive and splitting tensile strengths of the specimens, the result of the microscopic parameter analysis of scanning electron microscopy indicates that the performance of the steel fiber content for RPC is improved. In combination with the cost of RPC in the test, the cost–performance ratio of RPC at this stage was compared. The results provide a reference for the extensive use of RPC in practical engineering in the future.

### 4.1. Mechanical Test Failure Pattern of Fiber-Reinforced RPC

In the early stages of the compression failure test, the RPC matrix without fiber was used with the load gradually increasing. The concrete surface around the matrix fell off and produced debris. When the load reached approximately 70%–80% of the failure load, the fall of debris was frequent. When the failure load was reached, the specimen produced a huge bursting sound with the release of energy. With the addition of steel fiber, the bursting phenomenon under standard and natural curing conditions disappeared. The failure pattern of concrete specimens no longer achieved complete fragmentation, but exhibited improved integrity. When the fiber content is low (steel fiber contents of 1% and 2%), a large burst sound accompanied by obvious cracks on the surface of the specimen is still observed in the specimens involved in thermal curing. With the increase in fiber content, the crack sound in the destruction disappeared. The compound curing accelerated the hydration reaction of cement and the volcanic ash reaction of mineral admixture in the specimen. This reaction was more in-depth than those of standard and natural curing. The matrix strength was high, and the energy generated by the destruction of the specimen was large. Thus, minimal fiber content could not reduce the burst sound phenomenon generated by the energy release.

Figure 8 demonstrates that RPC with steel fiber content of 0% had a short failure time—from the beginning of the splitting tensile test to formation of the specimen. The specimen cracked into two parts under the load action. With the increase of fiber content, the specimen in the load process was uprooted. The fracture component increased with the decrease in steel fiber content. The specimen was not cut after the damage, but showed significant cracks in a relatively complete matrix.

### 4.2. Mechanical Test Results and Analysis of Fiber-Reinforced RPC

#### 4.2.1. Test Results

The mix ratio in Table 12 was adopted for standard and natural curing, in accordance with the comparison and analytical results of the basic mix ratio test of the orthogonal test above. The mix ratio in Table 13 was adopted for compound curing. The steel fiber content changed from 0% to 5%. The specific parameters are shown in Table 12 and Table 13.

#### 4.2.2. Performance Analysis of Steel Fiber RPC

The workability index of RPC obviously decreased with increase in steel fiber content. When the fiber content was 5%, the RPC’s flow performance decreased. In the slump test, the steel fibers enabled concrete to become whole. Accordingly, the slump bucket was difficult to fill. When the bucket was pulled out, a “bucket” shape appeared (Figure 9). This shows that when the steel fiber content reaches 5%, the working performance of RPC at this time is low, which is not conducive in practical engineering. In the mixture ratio of thermal curing conditions during the test, the water/binder ratio was 0.16, because the water glue was found to be low when the doped fiber increased. It was difficult to guarantee good work performance, even when the dosage of the water-reducing agent was controlled and a slurry is produced. The RPC slurry had a large amount of internal friction during the steel fiber distribution. The steel fiber occupied part of the free water and increased the internal friction of the slurry, thereby resulting in a dramatic decrease in RPC fluidity. The workability of the compound curing mix ratio with high steel fiber content was significantly reduced in comparison with the natural and standard curing mix ratios. Such a phenomenon was attributed to the low water/binder ratio in the compound curing mix ratio. This condition can affect compactness in the manufacturing process of components. The increase in internal defects can also adversely impact component strength.

#### 4.2.3. Compressive Strength of Steel Fiber RPC 

The compressive strength shown in Figure 10 and Figure 11 demonstrate that steel fiber content is 1%, 2%, 3%, 4%, and 5%. Under the different curing conditions, the compressive strength of the 28-day RPC increased by 33.14%, 24.46%, and 31.97% for standard, natural, and compound curing, respectively.

The RPC strength results of various ages under different curing conditions showed that steel fiber positively influences concrete. Seven days before maintenance, when the steel fiber content was 4%, the growth ratio of the compressive and tensile strengths of RPC reached maximum. Hot water curing promotes concrete hydration reaction and ash, and its hydration products make the matrix inside close-grained. The bonding between steel fibers became increasingly dense. Figure 11a demonstrates that RPC compressive strength with the 7-day compound curing rate increased more than those obtained by the other two maintenance methods. When the temperature drops and tends to be stable, the reaction in the RPC matrix slows down. The specimen strength under the standard curing and natural curing conditions gradually increased. However, the compressive strength of the matrix under the three curing conditions increased at a similar rate and stabilized with the progress of the reaction. RPC performance with steel fiber was found to be more stable than that without. Few mutations were observed. This result is helpful for practical engineering applications.

#### 4.2.4. Splitting Tensile Strength of Steel Fiber RPC

The splitting tensile strength shown in Figure 12 and Figure 13 demonstrated that steel fiber content is 1%, 2%, 3%, 4%, and 5%. The maximum tensile strength for 28-day curing increased by 148.95%, 185.97%, and 132.43%.

The test results demonstrated that steel fiber evidently affects the improvement of RPC tensile strength. The incorporation of steel fiber can disperse the shrinkage stress of a capillary tube in the matrix and effectively alleviate the local stress concentration phenomenon. The steel fiber bridge on both sides of tiny cracks are depicted in Figure 14a,b (fiber bonding micro-amplification). The steel fiber can bridge cracks in active powder concrete. Figure 14c exhibits that the exposed fiber surface is bound using spherical and flaky materials, thus covering the fiber surface and adhering to certain fragments. The RPC matrix is closely bound to the fiber. With the phase growth, the spherical silica fume attached to the fiber surface conducts a secondary hydration reaction with Ca(OH)_2_. The hydration product of cement was used to generate C–S–H gel with the network and dense structures that have strong chemical bonding and static friction forces with a rough fiber surface. The steel fiber surface is rough and uneven. Part of the matrix is embedded in the uneven surface of the steel fiber. When the fiber is pulled, the matrix squeezes the uneven fiber surface, thereby forming a strong grip force. The sliding friction generated by the slip of the fiber when it is pulled out forms the bonding force between the matrix and the steel fiber. Thus, steel fibers play a role of bridge and pin plug and hinder crack development. Figure 13 illustrates that RPC tensile performance is particularly significantly improved by steel fiber in comparison with compressive strength. The average increase was found to be 140%.

Steel fiber distribution is shown in Figure 14d. The figure demonstrates that fiber can penetrate the mixed matrix when the dosage of steel fiber is high. Matrix twisting and bending can occur in the steel fiber. The RPC slump is reduced with increase in steel fiber content; this slump has an effect on work performance and can be seen from the side. A high steel fiber content comprises of a part of free water. Consequently, the workability of the mixture worsens and will be unable to fully fill the micropores in the interface. This situation causes an increase in internal defects and leads to a decline in matrix strength. When the steel fiber content continues to increase, a steel fiber network is formed, resulting in the intercalation and mixing effect of the internal steel fiber. The fiber was found to lack sufficient slurry coating and filling. The bonding degree between fiber and RPC material decreases, and the friction decreases. This explains why the compressive strength and tensile strength of steel fibers no longer increase, but decrease, when the volume content of steel fibers exceeds 4%.

## 5. RPC Cost Analysis

Cost analysis of the RPC materials was conducted to facilitate their promotion and application. Table 14 illustrates the costs of various materials, ascertained using current prices in Changchun City, Jilin Province, China.

The test mix ratio cost calculation and cost performance ratio analysis are shown in Table 15.

The material cost performance is high when the fc/P and ft/P values are high. The difference between the performance–price ratio of ordinary C30 concrete and high-strength concrete C60, RPC, and steel fiber RPC has been narrowed down in Figure 15. The performance–price ratio of RPC decreased on the addition of steel fibers. Such an addition can improve RPC compressive strength. The cost was much higher for steel fiber RPC than C30 and C60 concrete due to the dimensional price factor. This resulted in a decrease in the performance–price ratio of steel fiber RPC with increase in steel fiber content. However, the RPC with steel fiber content of 1% under the compound curing condition had the same cost performance as C60. The RPC without steel fiber was better than that of the C60 concrete. From the perspective of cost–performance ratio of splitting tensile strength, the RPC concrete was found to be much higher than ordinary concrete. The cost–performance ratio of the tensile strength of steel fiber RPC gradually increased with increase in steel fiber content. The steel fiber RPC with optimal steel fiber content of 4% had a good cost–performance ratio of tensile strength. The RPC cost was based on the material procurement price experiment because the test quantity was small. The material purchasing price was close to that of the market retail price. The raw material purchase price will relatively decrease in large-scale batch production. Cost performance will also further reduce in the RPC, relative to the C30 and C60 concrete. This is conducive for the promotion and practical applications of RPC.

## 6. Conclusions

In this study, RPC was created under natural, standard, and compound curing conditions. The RPC’s mechanical properties and design method of steel fiber content were studied on the basis of compressive and splitting tensile strength tests. The following conclusions were drawn:The optimal mix ratio varied on the basis of the curing conditions. Under the standard and natural curing conditions, the water/binder ratio was 0.18, and the sand/binder ratio was 0.7; in addition, the volume content of fly ash was 20%, and the volume content of silica fume was 13%. The optimal water/binder ratio for compound curing was 0.16, and the sand/binder ratio was 0.7; moreover, the volume content of fly ash was 10%, and the volume content of silica fume was 13%.The different curing conditions significantly influenced the mechanical properties of active powder concrete for 7 days/28 days. During combined hot water curing, the strength of active powder concrete significantly increased in 7 days. This outcome was basically the same as that of 28 days and higher than that of 7 days/28 days under standard and natural curing conditions. Under the standard and natural curing conditions, the tensile and compressive strengths of the RPC specimens containing steel fibers were slightly different than those of hot-water curing. The result verified that an appropriate amount of steel fibers could effectively improve the internal structure of the matrix and stabilize the RPC performance. Choosing the appropriate heat curing temperature and time is beneficial for improving the strength of active powder concrete. Such an approach can reduce the cost of a prefabricated structure and accelerate the construction process.When the steel fiber content increased from 0% to 4%, the tensile and compressive strengths of steel fiber-reinforced RPC increased. The strength began to decline when the steel fiber content was 5%, thereby indicating that optimal content of the 12-mm steel fiber was 4%. Under standard curing, the compressive and tensile strengths reached 123.59 and 27.6 MPa, correspondingly. The compressive and tensile strengths reached 113.13 and 24.67 MPa, respectively, under natural curing. Under compound curing, the compressive and tensile strengths reached 140.14 and 29.6 MPa, correspondingly.The results of the microscopic structural analysis indicate that, on a micro level, the mixed short steel fiber can effectively improve the internal structure of the RPC, and the strength of RPC remains stable. Considerable steel fiber could cause internal fiber clustering, twisting, or bending, and weaken the internal area. Therefore, the optimal steel fiber dosage of the steel fiber-reinforced RPC is particularly important.At present, the cost of RPC is relatively expensive in comparison with that of ordinary concrete. However, the precast and compound forms of some components can be adopted. Such a strategy can not only reduce the consumption of RPC and engineering costs, but also ensure that the performance of the key parts and components can meet design requirements. From the perspective of development, the application of active powder concrete can effectively reduce carbon dioxide emission, thus indicating that active powder concrete is conducive for environmental protection.

## Figures and Tables

**Figure 1 materials-12-03615-f001:**
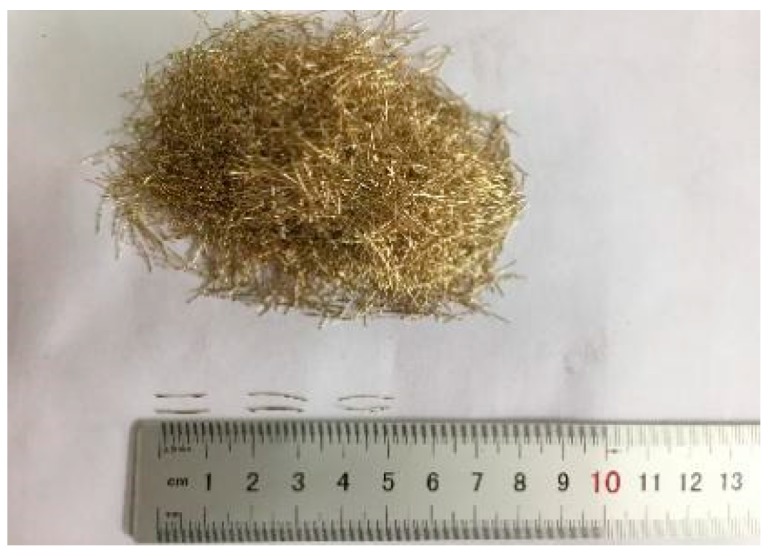
Steel fiber.

**Figure 2 materials-12-03615-f002:**
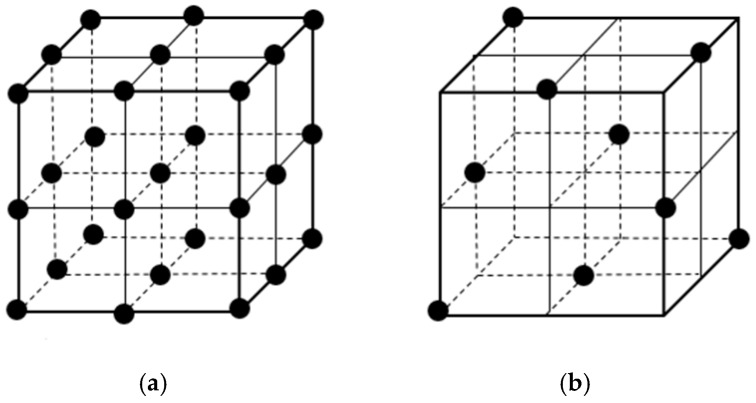
A four-factor layout: (**a**) comprehensive test; (**b**) orthogonal experimental design.

**Figure 3 materials-12-03615-f003:**
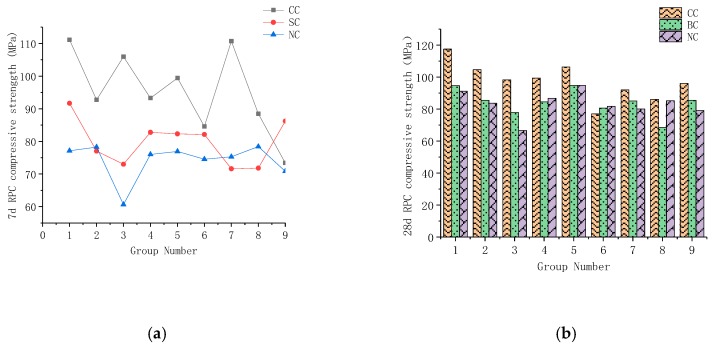
RPC compressive strength: (**a**) 7-day compressive strength; (**b**) 28-day compressive strength.

**Figure 4 materials-12-03615-f004:**
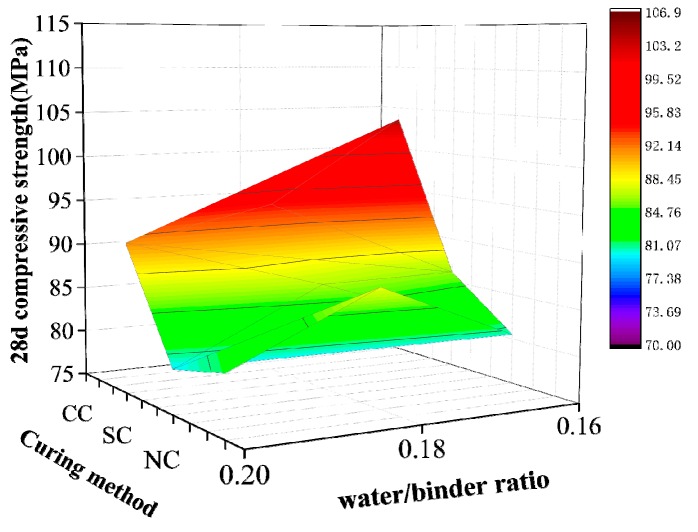
Water/binder ratio.

**Figure 5 materials-12-03615-f005:**
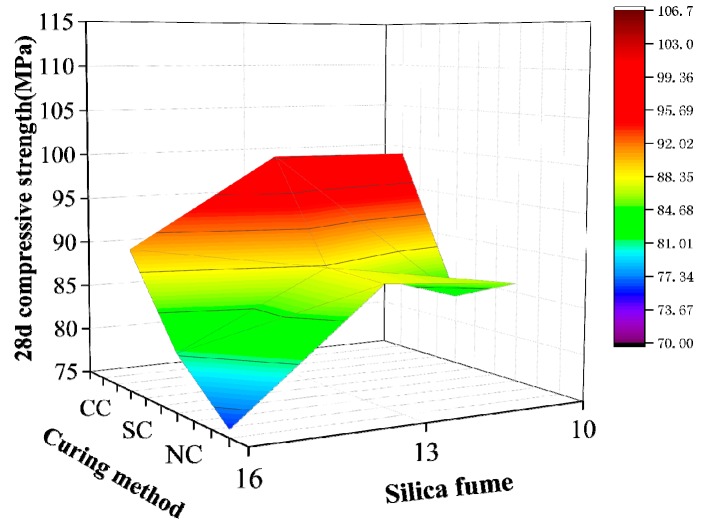
Silica fume content.

**Figure 6 materials-12-03615-f006:**
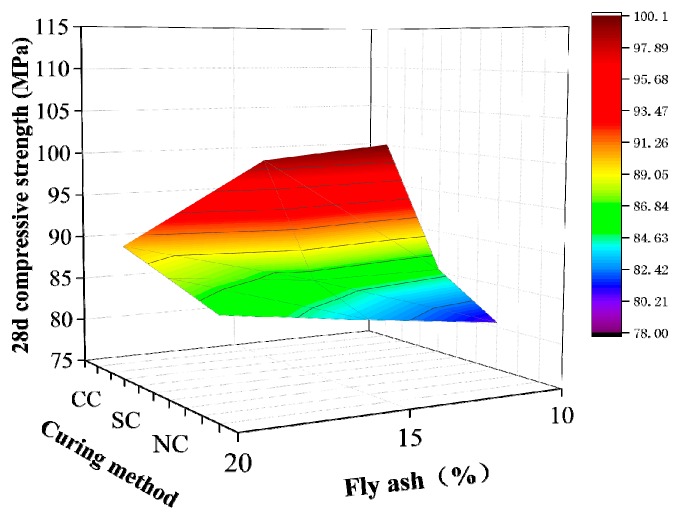
Fly ash content.

**Figure 7 materials-12-03615-f007:**
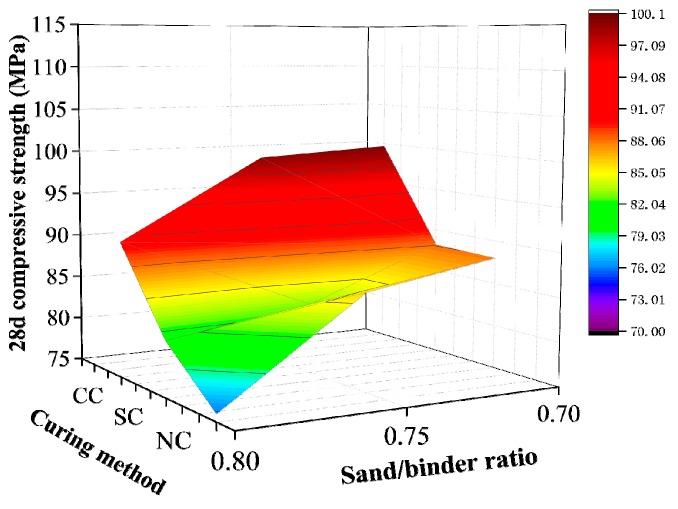
Sand/binder ratio.

**Figure 8 materials-12-03615-f008:**
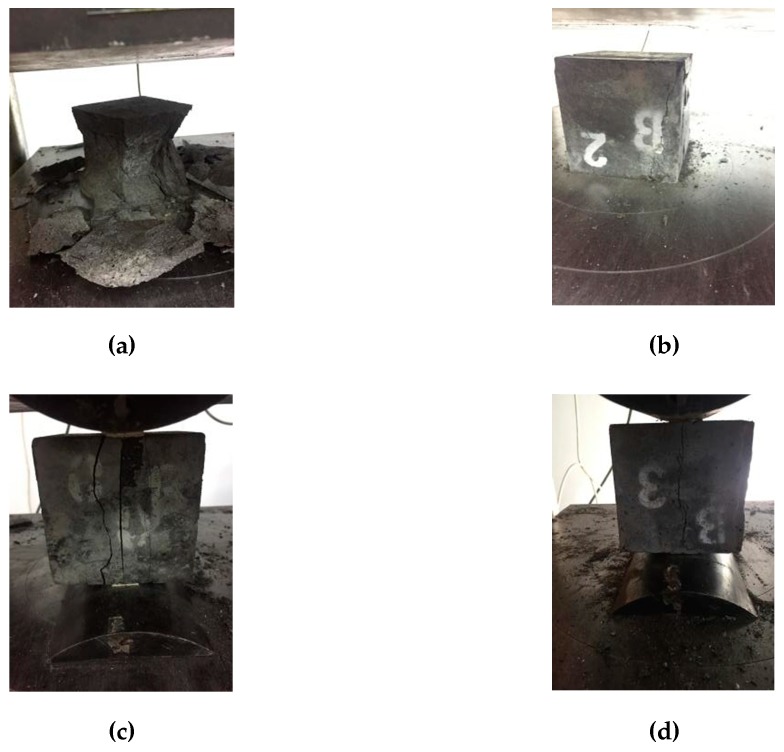
Influence of steel fiber on the mechanical properties of RPC: (**a**) compressive damage without steel fiber; (**b**) compressive damage with steel fiber; (**c**) split tensile failure without steel fiber; (**d**) split tensile failure with steel fiber.

**Figure 9 materials-12-03615-f009:**
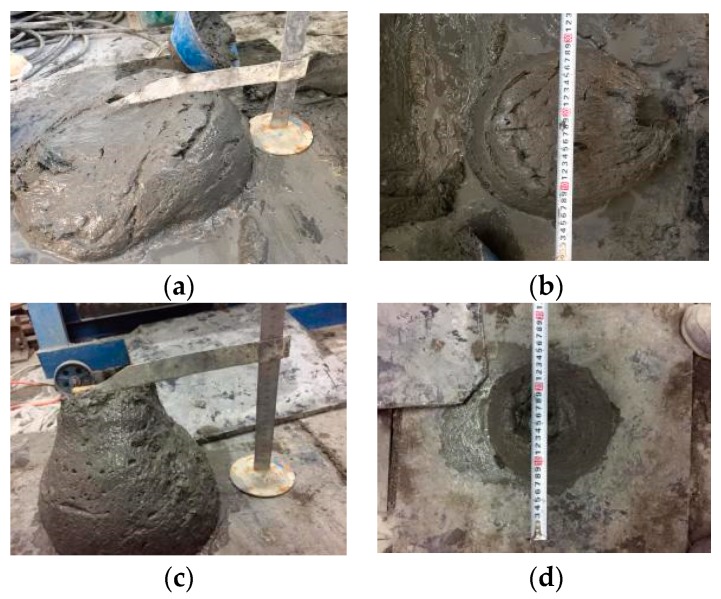
Comparison of RPC workability with different steel fiber dosages: (**a**) steel fiber content of 2%; (**b**) steel fiber content of 2%; (**c**) steel fiber content of 5%; (**d**) steel fiber content of 5%.

**Figure 10 materials-12-03615-f010:**
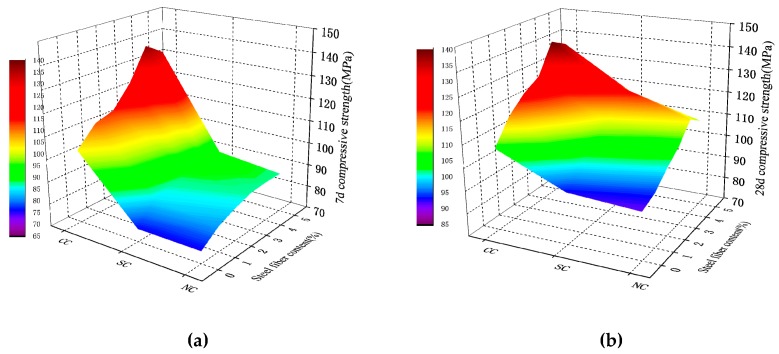
Compressive strength of steel fiber RPC with different dosages: (**a**) 7-day and (**b**) 28-day compressive strength.

**Figure 11 materials-12-03615-f011:**
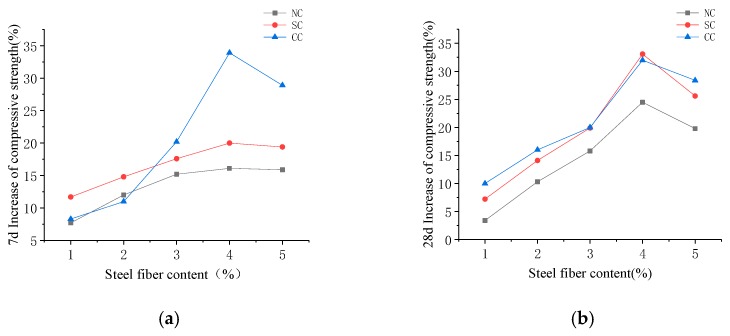
Percentage of compressive strength improvement of steel fiber RPC with different dosages of steel fiber after (**a**) 7-day curing and (**b**) 28-day curing.

**Figure 12 materials-12-03615-f012:**
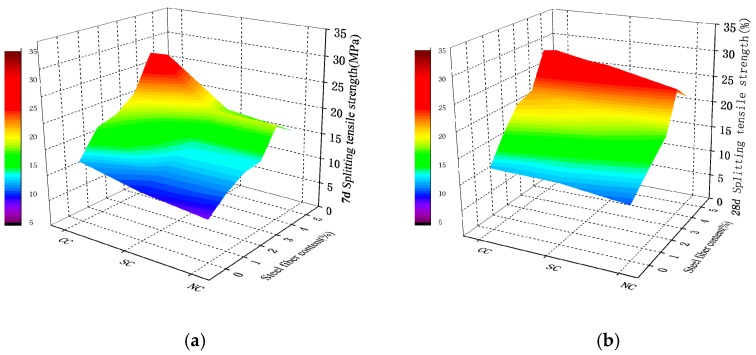
Splitting tensile strength of steel fiber RPC with different dosages: (**a**) 7-day splitting tensile strength; (**b**) 28-day splitting tensile strength.

**Figure 13 materials-12-03615-f013:**
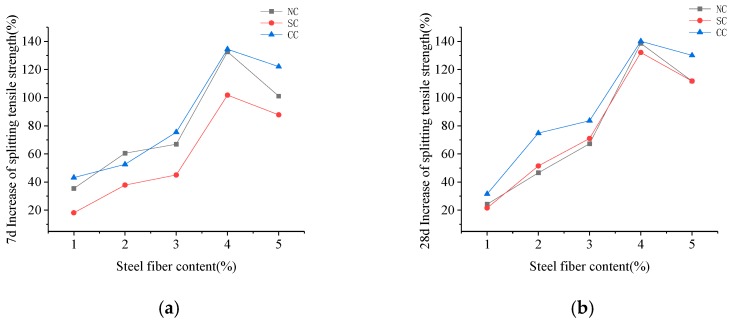
Percentage of splitting tensile strength improvement of steel fiber RPC with different dosages of steel fiber after (**a**) 7-day curing and (**b**) 28-day curing.

**Figure 14 materials-12-03615-f014:**
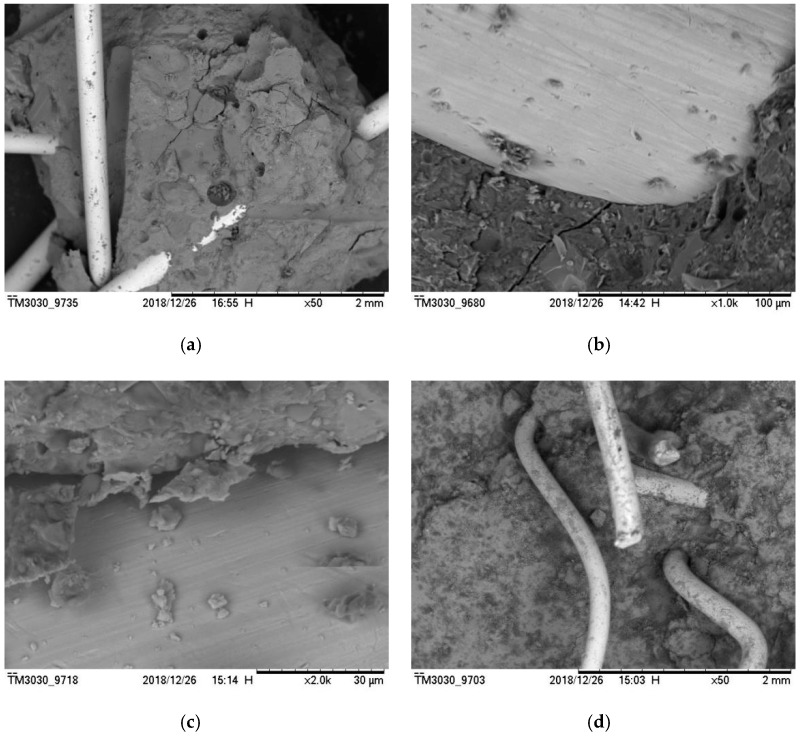
Micrograph of steel fiber-reinforced active powder concrete: (**a**) Steel fiber distribution; (**b**) steel fibers and tiny cracks; (**c**) steel fibers and hydration products; (**d**) the steel fibers clumped together.

**Figure 15 materials-12-03615-f015:**
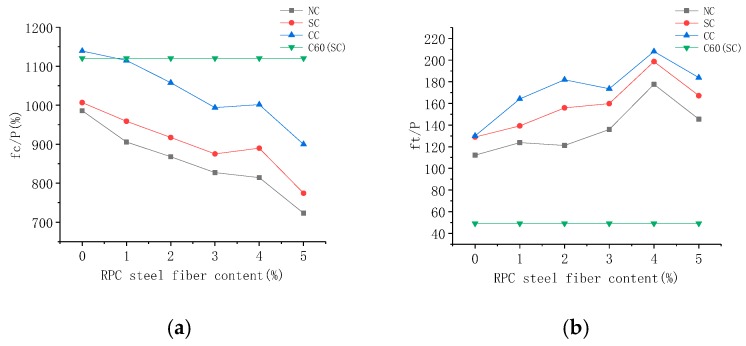
Cost performance analysis of steel fiber-reinforced active powder concrete and C60: (**a**) Compression strength cost performance comparison; (**b**) splitting tensile strength cost performance comparison.

**Table 1 materials-12-03615-t001:** Physical and chemical properties of cement.

Properties		Standard Value	Actual Value
Physical properties	Specific surface area (m^2^ /kg)	≥300	367
	Initial set (min)	≥45	99
	Final set (min)	≤390	145
Compressive strength	3 day (MPa)	≥23.0	29.1
	7 day (MPa)	≥52.5	
Flexural strength	3 day (MPa)	≥4.0	6.0
	7 day (MPa)	≥7.0	
Chemical properties	Stability	Qualified	Qualified
	Loss on ignition (%)	≤3.5	1.61
	MgO (%)	≤5.0	0.98
	SO_3_ (%)	≤3.5	2.62
	Insolubles (%)	≤1.5	1.01
	Cl^−^(%)	≤0.06	0.007

**Table 2 materials-12-03615-t002:** Physical and chemical properties of silica fume.

Properties		Standard Value	Actual Value
Physical properties	Specific surface area (m^2^ /kg)	≥15	20
	Pozzolanic activity index (%)	≥15	116
Chemical properties	SiO_2_ (%)	≥15	94.5
	Loss on ignition (%)	≤6.0	2.5
	Cl^−^(%)	≤0.02	0.02
	Moisture content (%)	≤3.0	1.2
	Water demand ratio (%)	≤125	118

**Table 3 materials-12-03615-t003:** Chemical properties of fly ash.

Properties		Standard Value	Actual Value
Chemical properties	Water demand ratio (%)	≤105	94
	Loss on ignition (%)	≤8.0	1.0
	Moisture content (%)	≤1.0	0.1
	SO_3_ (%)	≤3.0	0.3
	CaO_3_ (%)	≤1.0	0.16
	MgO (%)	≤5.0	1.08
	Cl^−^ (%)	≤0.02	0.01

**Table 4 materials-12-03615-t004:** Physical and chemical properties of mineral powder.

Properties			Standard Value	Actual Value
Physical properties	Specific surface area (m^2^ /kg)		≥400	429
	Liquidity ratio (%)		≥95	102
	Density (%)		≥2.8	2.9
Chemical properties	Moisture content (%)		≤1.0	0.1
	Loss on ignition (%)		≤3.0	1.07
	Pozzolanic activity index (%)	7 day	≥75	83
		28 day	≥95	98

**Table 5 materials-12-03615-t005:** Physical properties of steel fiber.

Index	Diameter/mm	Length/mm	Aspect Ratio	Tensile Strength/MPa
Unit value	0.2	12	60	2850

**Table 6 materials-12-03615-t006:** Maintenance methods.

Maintenance Method		Specific Process
NC	Natural curing	Indoor natural curing: 1-day demolding and maintained to 7 and 28 days under indoor natural curing conditions.
SC	Standard curing	Standard curing: 1-day demolding and maintained to 7 and 28 days under standard curing conditions.
CC	compound curing	Indoor natural curing: 1-day demolding, hot water (60 ± 1 °C) for 36 hours, and maintained to 7 and 28 days under indoor natural curing conditions.

**Table 7 materials-12-03615-t007:** Factor level.

Factors	Units	Levels		
		1	2	3
Water/binder ratio	-	0.16	0.18	0.20
Sand/binder ratio	-	0.70	0.75	0.80
Silica fume	%	10	13	16
Fly ash	%	10	15	20

**Table 8 materials-12-03615-t008:** Orthogonal test mix ratio.

No.	Gelled Material	Water/Binder Ratio	Cement (%)	Silica Fume (%)	Fly Ash (%)	Mineral Powder (%)	Sand/Binder Ratio	Superplasticizer (%)
B1	1400	0.16	75	10	10	5	0.70	1.5
B2	1400	0.16	67	13	15	5	0.75	1.5
B3	1400	0.16	59	16	20	5	0.80	1.5
B4	1400	0.18	70	10	15	5	0.80	1.5
B5	1400	0.18	62	13	20	5	0.70	1.5
B6	1400	0.18	69	16	10	5	0.75	1.5
B7	1400	0.20	65	10	20	5	0.75	1.5
B8	1400	0.20	72	13	10	5	0.80	1.5
B9	1400	0.20	64	16	15	5	0.70	1.5

**Table 9 materials-12-03615-t009:** Compressive strength range analysis.

Maintenance Method	Factors	1	2	3	R
CC	water/binder ratio	103.52	94.280	91.400	12.12
	silica fume	99.07	99.65	90.48	9.67
	fly ash	100.050	98.963	90.19	9.86
	sand/binder ratio	99.65	99.07	90.477	9.17
SC	water/binder ratio	86.04	86.52	79.75	6.77
	silica fume	82.95	88.16	81.2	6.96
	fly ash	85.25	85.92	88.16	2.91
	sand/binder ratio	88.157	82.940	81.250	6.96
NC	water/binder ratio	80.50	87.81	81.52	7.31
	silica fume	86.04	87.95	75.84	12.11
	fly ash	80.56	83.203	86.06	5.50
	sand/binder ratio	87.95	86.04	75.84	12.11

**Table 10 materials-12-03615-t010:** Range analysis of the ratio of factors in the optimal group.

Factors	Units	Maintenance Method		
		SC	NC	CC
Water/binder ratio	–	0.18	0.18	0.16
Sand/binder ratio	–	0.70	0.70	0.70
Silica fume	%	13	13	13
Fly ash	%	20	20	10

**Table 11 materials-12-03615-t011:** Mechanical properties of the optimum mixing ratio for range analysis.

Curing Method	Preparation Parameters				Work Performance		Compressive Strength	
	Water/Binder Ratio	Silica Fume (%)	Fly Ash (%)	Sand/Binder Ratio	Extension Degree/mm	Slump/mm	7 Day/MPa	28 Day/MPa
CC	0.16	13	10	0.7	280	200	103.54	106.19
SC	0.18	13	20	0.7	325	230	80.76	94.82
NC	0.18	13	20	0.7	325	240	78.67	91.90

**Table 12 materials-12-03615-t012:** Mechanical properties of steel fiber-reinforced RPC under standard and natural curing conditions.

No.	RPC Mix					Work Performance		28 Day SC (MPa)		28 Day NC (MPa)	
	Steel Fiber Content (%)	Water/ Binder ratio	Silica Fume (%)	Fly Ash (%)	Sand/ Binder Ratio	Extension Degree/ mm	Slump/ mm	Compressive Strength	Splitting Tensile Strength	Compressive Strength	Splitting Tensile Strength
B0	0	0.18	13	20	0.70	360	260	92.83	11.89	90.90	10.34
B1	1	0.18	13	20	0.70	260	230	99.54	14.46	94.00	12.85
B2	2	0.18	13	20	0.70	235	210	105.94	18.01	100.23	15.16
B3	3	0.18	13	20	0.70	240	210	111.32	20.33	105.22	17.30
B4	4	0.18	13	20	0.70	240	200	123.59	27.60	113.13	24.67
B5	5	0.18	13	20	0.70	170	100	116.55	25.18	108.90	21.90

**Table 13 materials-12-03615-t013:** Mechanical properties of steel fiber-reinforced RPC under compound curing conditions.

No.	RPC Mix					Work Performance		28 day CC (MPa)	
	Steel Fiber Content (%)	Water/ Binder Ratio	Silica Fume (%)	Fly Ash (%)	Sand/ Binder Ratio	Extension Degree/mm	Slump/ mm	Compressive Strength	Splitting Tensile Strength
B00	0	0.16	13	10	0.70	260	220	106.19	12.12
B11	1	0.16	13	10	0.70	245	200	116.86	17.21
B22	2	0.16	13	10	0.70	210	190	123.23	21.19
B33	3	0.16	13	10	0.70	195	180	127.40	22.25
B44	4	0.16	13	10	0.70	185	150	140.14	29.10
B55	5	0.16	13	10	0.70	155	110	136.40	27.87

**Table 14 materials-12-03615-t014:** Test material price information.

Cement	Mineral Powder	Fly Ash	Silica Fume	River Sand	Water Reducer	Steel Fiber	Gravel
0.51	0.05	0.08	1.22	0.03	9.6	1.5	0.06

Note: The unit of price in the table is yuan/kg.

**Table 15 materials-12-03615-t015:** RPC cost performance analysis.

Concrete Label	Maintenance Method	fc/MPa	ft/MPa	P/(Ten Thousand Yuan/kg)	fc/P	ft/P
C30	SC	35	2.01	0.038	921.053	52.895
C60	SC	65	2.85	0.058	1120.690	49.138
RPC Steel fiber content (0%)	NC	90.9	10.34	0.0922	985.900	112.148
	SC	92.83	11.89	0.0922	1006.833	128.959
	CC	106.19	12.12	0.0932	1139.378	130.043
RPC Steel fiber content (1%)	NC	94	12.85	0.1038	905.588	123.796
	SC	99.54	14.46	0.1038	958.960	139.306
	CC	116.86	17.21	0.1048	1115.076	164.218
RPC Steel fiber content (2%)	NC	100.23	15.16	0.1155	867.792	131.255
	SC	105.94	18.01	0.1155	917.229	155.931
	CC	123.23	21.19	0.1165	1057.768	181.888
RPC Steel fiber content (3%)	NC	105.22	17.3	0.1272	827.201	136.006
	SC	111.32	20.33	0.1272	875.157	159.827
	CC	127.4	22.25	0.1282	993.760	173.557
RPC Steel fiber content (4%)	NC	113.13	24.67	0.1389	814.471	177.610
	SC	123.59	27.6	0.1389	889.777	198.704
	CC	140.14	29.1	0.1399	1001.716	208.006
RPC Steel fiber content (5%)	NC	108.9	21.9	0.1506	723.108	145.418
	SC	116.59	25.18	0.1506	774.170	167.198
	CC	136.4	27.87	0.1516	899.736	183.839

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
