# Peer review of "Reactive Powder Concrete Mix Ratio and Steel Fiber Content Optimization under Different Curing Conditions"

_materials, 2019, doi:10.3390/ma12213615_

Round 1
Reviewer 1 Report
The study evaluates the improvement effect of various steel fiber contents on the compressive performance of the reactive powder concrete. The results are interesting and provides potential for practical use. As the technical quality of the manuscript could be further improved (to enhance the readability), my overall recommendation is Minor revision prior to its publication.
Specific comments and suggestions
Title: “RPC Mix Ratio …” Is the abbreviation “RPC” common enough to be clear even for the researchers not working exactly in this research field?
Please avoid using multiple references. E.g., lines 59-61: RPC performance is directly related to the fibre content [11-33] …” and mention their contribution/significance separately.
At the end of Introduction, a new paragraph should be added indicating the objectives of the study (connected with knowledge gaps described within introduction).
Table 1-5 should be checked carefully for typographical and other mistakes.
Tables 1-5. Please check and correct all upper and lower indexes (e.g., m2, Cl-, SiO2, CaCO3 etc.)
Table 5: i) Kindly check the formula NaSO4 and ii) Is it really “Bleeding rate” that is tabulated? Or “Blending rate”?
Table 3: Please correct CaO3
Lines 120-121: References in the text – “[6-15, 18-33]”. Please avoid using multiple references and mention the contribution of each reference separately.
Fig. 5 caption: Why 3 absolutely the same texts are used there (“28 d RPC compressive strength”)? And also, the description of the horizontal axis (Fig. 5a) is not in English.
Line 173: In my opinion, instead of “Figures 5-8”, “Figures 6-9” should be used.
Line 167: “3.2.1. Analysis of range”. The correct section number is 3.2.2. (as 3.2.1. has already been used before).
Author Response
Reply to the Review Report
The author has made modifications according to the review opinions.Here are the changes:
In summary, because there are many factors affecting the matrix strength of RPC materials.Therefore,the present study used conventional materials and compressive strength consequence of orthogonal test to determine the optimal basic reference ratio of RPC under the following four factors: water/binder ratio, silica fume volume content, sand/binder ratio, and fly ash volume content. On this basis, steel fibers with a volume of 1%–5% were added to the RPC, and the five-volume steel fibers were subjected to basic curing tests for 7 and 28 d under natural, standard, and composite curing conditions. The compressive and splitting tensile strengths were studied, and the microscopic recorded data of RPC were analyzed by using scanning electron microscopy.

Reviewer 2 Report
The author conducted rigorous tests on reactive powder concrete and the collected data are enough to be recommended for publication. However, the manuscript is poorly written in terms of organization, analyses of results and introduction of concepts. Hence, it is required to go through a major revision as per the following comments/recommendations.
The abstract does not mention the analytical method used to optimize the mixing ratios. In addition, the novelty is not at all clear.
The paper needs to be organized as follows (A, B and C):
(A) IN THE INTRODUCTION, PLEASE CONSIDER THE FOLLOWING:
Introduce what RPC is? How is it different from conventional concrete? What are the applications of use? Define the difference between the natural, standard and compound curing techniques Provide enough reference for the addition of effects of adding steel fibres to concrete (lines 61-65) Introduce what is the orthogonal test and how will it be able to optimize the mix Identify the gaps in the existing literature
(B) Materials and Methods:
Figure 2 is unneeded Figure 3 is unneeded In 2.2: explain the difference between “indoor” and “outdoor” environments in more details. In lines 120-130, please re-write this paragraph in a way that every sentence is ended by the reference from which it was taken instead of the current state of all the references being in the first line. After that, move the whole paragraph to the introduction section. Concerning references, as a general comment., most of the papers cited are in Chinese as seen in the bibliography. It is advised to replace most of these references with ones that are in English so that the reader can be able to trace the arguments back to the original references. It seems to be like a cultural, the Chinese authors only consider the studies of their country. Please, consider other continents. Add section 2.3 titled “testing program”. In this section, introduce in details the standards, equipment and procedure of all the tests that were done on the RPC which were later explained in the results such as strength test, slump and SEM. Add section 2.4 titled “Optimizing RPC mix”. In this section, introduce the process of using the orthogonal test to identify the significant parameters in the mix design. This way the reader can understand the results that follow. In this section, a table is needed in which the mixing proportions of the mixes which were tested is written. Otherwise, the results are not reproducible.(C) Results and discussions (not mechanical properties of RPC)
Section 3.1 would be the results for the mechanical properties of the mixes tested. The comparisons are done based on the selected ratios and curing techniques. While discussing the failure shapes and patterns, do not mentioned discussions that are not quantifiable such as the sound of cracking of the specimen. In lines 194-201, please add references to the discussions. In line 196, Is the volcanic ash meant to be Fly ash? Please be consistent. Section 3.2 would be the optimization results in which the orthogonal test is done using the results from 3.1 and the results of the most significant parameters in the mix design are concluded Section 3.3 would be the results for the fibre reinforced concrete samples. In line 282, what is meant by mobility? In line 284, why is it significant to say “bucket” shape? The slump test is quantitative not qualitative In line 288, Is the RPC produced a slurry? In figure 12 and 14, the coloured axes should be labelled. What do they represent? In line 319, what is meant by jumps? The discussion of the fibre reinforced concrete results is full of logical leaps. Please re-write it focusing only on the results obtained in this study. For example, in lines 328-331, the enhanced performance of the concrete containing steel fibres as opposed to the control is explained using generic terms such as alleviating the local stress concentration which is not backed up by any microstructural modelling or testing. For the results of the SEM, it should not be a section on its own, rather, the results should be used as evidence to support certain claims deduced from the results in previous sections. Anyway, the SEM images in figure 16 need to have a scale, magnification factor and image sharpness indicator.
SYNTAX ERRORS TO BE CONSIDERED:
Line 12: consider changing the word “realistic” Line 24: What is meant by “later lifting”? Line 83: change method to methods Table 1: change specifific to specific, qualifified to qualified and SO3 to SO3 Table 2: change m2 to m2, SO2 to SO2 and specifific to specific Table 3: change specifific to specific Table 4: change NaSO4 to NaSO4 Table 5 is not required if the superplasticizer used is a commercial product Table 6 is said in the text to contain fine aggregates specifications but instead it includes that of the fibres. Please be consistent. Figure 2 is not needed. Please remove. Figure 3 is not needed. Please remove. Line 118: change the word compound to composite Line 127: change the word ratio to ratios object to objective Figure 5: The x-axis is written in Chinese. Please correct. Line 319: consider changing the word jumps Line 364: change all the numbers in all chemical compounds to be subscripts Line 409: please reconsider the use of the word sexual
Author Response
The author has modified according to your suggestion.
Indoor:At the time of the test, the outdoor temperature was reduced to below -5℃, and considering that the application of materials should conform to the actual engineering situation, the environment and temperature of the test workshop were in line with the natural curing conditions, so indoor natural curing was adopted.
Introduce what RPC is? How is it different from conventional concrete? What are the applications of use?Reply:Therefore, the high-performance structure based on RPC material can greatly save the amount of material, reduce the size of the structure, make the structure lighter, reduce energy consumption and pollution, and improve the long durability and the economy of the whole life of the structure. RPC structure can greatly simplify or even completely eliminate ordinary steel reinforcement, simplify construction process, be suitable for assembling construction, reduce cost, and hopefully solve the problems existing in existing Bridges, such as fatigue cracking of bridge deck, fragile pavement, cracking and bending of traditional prestressed concrete beams, low assembling rate of small and medium-sized span fabricated Bridges, etc.
Define the difference between the natural, standard and compound curing techniques
Reply:The curing temperature and humidity of standard curing are maintained in a constant state, which is conducive to the formation of RPC body strength. However, the temperature of natural curing environment cannot be maintained stable, and the humidity of air is far less than that of standard curing. Compound curing is the heat curing of a period of time under the foundation of natural curing.
Provide enough reference for the addition of effects of adding steel fibres to concrete (lines 61-65)
Reply:Modified as required
Introduce what is the orthogonal test and how will it be able to optimize the mix
Reply:Orthogonal experimental is a kind of test method that can select representative test points from comprehensive test in an efficient and rapid way. The selected test points are "evenly dispersed and in comparable order". Therefore, when dealing with multi-factor and multi-level problems, as shown in figure 2, samples selected from the orthogonal experiment have uniform dispersion degree. According to the characteristics of this experiment, the number of experiments can be reduced to the greatest extent, and the results are conducive to the scientific judgment and analysis of the experimental results.
|
(a) |
(b) |
Figure 2. A four-factor layout: (a) comprehensive test; (b) orthogonal experimental design.
In summary, because there are many factors affecting the matrix strength of RPC materials.Therefore,the present study used conventional materials and compressive strength consequence of orthogonal test to determine the optimal basic reference ratio of RPC under the following four factors: water/binder ratio, silica fume volume content, sand/binder ratio, and fly ash volume content. On this basis, steel fibers with a volume of 1%–5% were added to the RPC, and the five-volume steel fibers were subjected to basic curing tests for 7 and 28 d under natural, standard, and composite curing conditions. The compressive and splitting tensile strengths were studied, and the microscopic recorded data of RPC were analyzed by using scanning electron microscopy.
Identify the gaps in the existing literature
Reply:There is a very direct relationship between material and fit ratio of RPC matrix strength. Compared with ordinary concrete, RPC based on maximum compactness theory has better aggregate gradation. Furthermore, the use of mineral admixture can not only effectively reduce the hydration heat of cement, but also effectively enhance the matrix strength of RPC on the premise of reducing the amount of cement. Due to the application of nanotechnology, many nanoscale materials have been proved to be helpful for the properties of RPC materials. At present, there are many studies on nano silica [10, 11], nano calcium carbonate [12], carbon nanotubes [13-15], metakaolin [16, 17], rice husk ash [18], glass powder [19], and Superfine cement and mineral powder. The results of the study on the mixture ratio are not the same,but most of them focus on one or more of the factors such as water-binder ratio [20-25], sand-binder ratio [21-24], fiber content [21-25], fly ash content [22,23,25]and silica fume content [21-23,25]. With high compressive strength and tensile strength as the target, according to the analysis of the mixing ratio in the current literature, it can be seen that the proportion of steel fiber content and mineral admixture on the compressive strength, especially the tensile strength, of RPC matrix is very significant. In addition, the water/binder ratio and sand/binder ratio that affect the basic properties of concrete also account for a large proportion. When these nanoscale admixtures enhance the strength of RPC matrix, the cost of RPC also increases, which is not conducive to the promotion and application of RPC materials in practical projects. How to use common raw materials to prepare RPC with excellent performance is the focus of current work, so the key to prepare efficient RPC is to need the ratio of various materials.
Figure 2 is unneeded
Reply:Deleted as requested
Figure 3 is unneeded
Reply:Deleted as requested
In 2.2: explain the difference between “indoor” and “outdoor” environments in more details.
Reply:Indoor:At the time of the test, the outdoor temperature was reduced to below
-5℃, and considering that the application of materials should conform to the actual engineering situation, the environment and temperature of the test workshop were in line with the natural curing conditions, so indoor natural curing was adopted.
In lines 120-130, please re-write this paragraph in a way that every sentence is ended by the reference from which it was taken instead of the current state of all the references being in the first line. After that, move the whole paragraph to the introduction section.
Reply:Modified as required
Concerning references, as a general comment., most of the papers cited are in Chinese as seen in the bibliography. It is advised to replace most of these references with ones that are in English so that the reader can be able to trace the arguments back to the original references. It seems to be like a cultural, the Chinese authors only consider the studies of their country. Please, consider other continents.
Reply:Modified as required
Add section 2.3 titled “testing program”. In this section, introduce in details the standards, equipment and procedure of all the tests that were done on the RPC which were later explained in the results such as strength test, slump and SEM.
Reply:Modified as required
Add section 2.4 titled “Optimizing RPC mix”. In this section, introduce the process of using the orthogonal test to identify the significant parameters in the mix design. This way the reader can understand the results that follow.
Reply:Modified as required
In this section, a table is needed in which the mixing proportions of the mixes which were tested is written. Otherwise, the results are not reproducible.
Reply:Modified as required
Section 3.1 would be the results for the mechanical properties of the mixes tested. The comparisons are done based on the selected ratios and curing techniques.
Reply:Modified as required
While discussing the failure shapes and patterns, do not mentioned discussions that are not quantifiable such as the sound of cracking of the specimen.
Reply:Modified as required
In lines 194-201, please add references to the discussions.
Reply:Modified as required
In line 196, Is the volcanic ash meant to be Fly ash? Please be consistent.
Reply:Volcanicc ash is not fly ash, but the volcanic reaction capability of fly ash itself
Section 3.2 would be the optimization results in which the orthogonal test is done using the results from 3.1 and the results of the most significant parameters in the mix design are concluded
Reply:Modified as required
Section 3.3 would be the results for the fibre reinforced concrete samples.
Reply:Modified as required
In line 282, what is meant by mobility?
Reply:Fluidity represents the performance of concrete
In line 284, why is it significant to say “bucket” shape? The slump test is quantitative not qualitative
Reply:The “bucket” shape: This shows that when the steel fiber content reaches 5%, the working performance of RPC at this time is very low, which is not conducive to the use of practical engineering.
In line 288, Is the RPC produced a slurry?
Reply:Produced a slurry
In figure 12 and 14, the coloured axes should be labelled. What do they represent?
Reply:The color axis represents the range of intensity values
In line 319, what is meant by jumps?
Reply:The“jumps”:The meaning of mutation
The discussion of the fibre reinforced concrete results is full of logical leaps. Please re-write it focusing only on the results obtained in this study. For example, in lines 328-331, the enhanced performance of the concrete containing steel fibres as opposed to the control is explained using generic terms such as alleviating the local stress concentration which is not backed up by any microstructural modelling or testing.
Reply:Modified as required
For the results of the SEM, it should not be a section on its own, rather, the results should be used as evidence to support certain claims deduced from the results in previous sections.
Reply:Modified as required
Anyway, the SEM images in figure 16 need to have a scale, magnification factor and image sharpness indicator.
Reply:Modified as required
Line 12: consider changing the word “realistic”
Reply:Modified as required
Line 24: What is meant by “later lifting”?
Reply:Modified as required
Line 83: change method to methods
Reply:Modified as required
Table 1: change specifific to specific, qualifified to qualified and SO3 to SO3
Reply:Modified as required
Table 2: change m2 to m2, SO2 to SO2 and specifific to specific
Reply:Modified as required
Table 3: change specifific to specific
Reply:Modified as required
Table 4: change NaSO4 to NaSO4
Reply:Modified as required
Table 5 is not required if the superplasticizer used is a commercial product
Reply:Deleted as requested
Table 6 is said in the text to contain fine aggregates specifications but instead it includes
that of the fibres. Please be consistent.
Reply:Modified as required
Figure 2 is not needed. Please remove.
Reply:Deleted as requested
Figure 3 is not needed. Please remove.
Reply:Deleted as requested
Line 118: change the word compound to composite
Reply:Modified as required
Line 127: change the word ratio to ratios object to objective
Reply:Modified as required
Figure 5: The x-axis is written in Chinese. Please correct.
Reply:Modified as required
Line 319: consider changing the word jumps
Reply:Modified as required
Line 364: change all the numbers in all chemical compounds to be subscripts
Reply:Modified as required
Line 409: please reconsider the use of the word sexual
Reply:Modified as required

Reviewer 3 Report
This paper investigates the reactive powder concrete mix ratio and steel fiber content optimization under different curing conditions. The novelty of work is poor and the paper is written in very confusing way. The depth of analysis is too low and difficult to follow the results. Too many tables should be written in a better way. Some results are presented in both table and figure. I can give my recommendation after an overall reconsideration. There are some comments that can help to improve the work:
-At the end of introduction section, you can explain the novelty of your research.
-There are three different names for curing in the hot water. In Fig.3 (c) is written hot water curing, in table 7 is written composite curing and in line 118, it is written compound curing. Please select one name and replace with others.
-In table 8, what do you mean by levels factor? Do you mean 3 different ratios were used in this research? Please explain it in a clear way.
-Pictures in Fig.4 are not presented anything special. Please explain in more scientific way or find better pictures to interpret your results.
-Different ratios should of w/b, SF and FA should not be written in Table 9. Please move those data to the experimental section.
-What is the difference between figure 8 a and b? Both figures present the 28d compressive strength. In Fig 8 a, the axis title should be written in English.
-How did you calculate the results in section 3.2.1? This section is not clear at all.
Author Response
Reply to the Review Report
The author has made modifications according to the review opinions.Here are the changes:
(1)Orthogonal experimental is a kind of test method that can select representative test points from comprehensive test in an efficient and rapid way. The selected test points are "evenly dispersed and in comparable order". Therefore, when dealing with multi-factor and multi-level problems, as shown in figure 2, samples selected from the orthogonal experiment have uniform dispersion degree. According to the characteristics of this experiment, the number of experiments can be reduced to the greatest extent, and the results are conducive to the scientific judgment and analysis of the experimental results.
|
|
|
|
(a) |
(b) |
Figure 2. A four-factor layout: (a) comprehensive test; (b) orthogonal experimental design.
(2)Calculation method of range analysis:
The main effect=(The maximum mean value of a test value at a leve)-(The minimum mean of all test values at a certain level)
(a)The sum of the observed values of each factor level was calculated respectively;
(b)Calculate the average value of each factor at each level;
(c)Calculate the effect of each factor, namely the maximum mean value of the same level minus the minimum mean value.

Round 2
Reviewer 2 Report
In terms of quality, the paper can now be recommended for publication. Since I am not a native English speaker, I cannot judge the English grammar. However, I suggest the authors to carefully re-read the manuscript in order to avoid any non-necessary expressions.
Therefore: I will consider a minor revision only for the authors to re-read the manuscript again.
Author Response
Thank you for your advice.
I have revised the English grammar of the manuscript according to your suggestion.

Reviewer 3 Report
However, the authors could improve the quality of the manuscript, but still, there are some issues that should be fixed:
-There are some English errors that should be fixed: Introduction section, first line, Reactive powder concrete (RPC) is "a" low energy consumption "and" eco-friendly cement-based...., or in section 2.2, Table 6 "presents" the specific process.... Please read the whole text again and correct all grammatical errors.
-In Table 6, please write NC next to each other, now C is shifted to the next line.
-In section 3.1, please present your data only in the table or figure. There are now the same 7d compressive strength and 28d compressive strength results in both Table 9 and Fig. 3. Please do the same for the rest of the article.
-If you want to keep Fig.3, please correct the vertical title of (a) and write the "7d RPC compressive strength".
-The caption for Fig. 8 is written 2 times, please correct it.
-In the caption of Table 14, it is written composite curing conditions, please change to the compound curing condition and check carefully through the manuscript to change the composite curing to the compound curing.
-The caption of Fig.9 should be written under the picture. Please correct it.
-The caption of Fig. 11 should be changed to the " Percentage of compressive strength improvement of steel fiber RPC with different dosages of steel fiber after (a) 7 d curing and (b) 28 d curing". Please do the same for caption of Fig.13.
-Please add a short explanation for each picture (a, b, c, and d) of Fig. 14 in caption and picture.
Author Response
1、There are some English errors that should be fixed: Introduction section, first line,
Reactive powder concrete (RPC) is "a" low energy consumption "and" eco-friendly
cement-based...., or in section 2.2, Table 6 "presents" the specific process.... Please read the
whole text again and correct all grammatical errors.
Reply:Modified as required
2、In Table 6, please write NC next to each other, now C is shifted to the next line.
Reply:Modified as required
3、In section 3.1, please present your data only in the table or figure. There are now the
same 7d compressive strength and 28d compressive strength results in both Table 9 and Fig. 3
Please do the same for the rest of the article.
Reply:Modified as required
4、If you want to keep Fig.3, please correct the vertical title of (a) and write the "7d RPC
compressive strength".
Reply:Modified as required
5、The caption for Fig. 8 is written 2 times, please correct it.
Reply:Modified as required
6、In the caption of Table 14, it is written composite curing conditions, please change to the
compound curing condition and check carefully through the manuscript to change the
composite curing to the compound curing.
Reply:Modified as required
7、The caption of Fig.9 should be written under the picture. Please correct it.
Reply:Modified as required
8、The caption of Fig. 11 should be changed to the " Percentage of compressive strength
improvement of steel fiber RPC with different dosages of steel fiber after (a) 7 d curing and (b) 28 d curing". Please do the same for caption of Fig.13.
Reply:Modified as required
9、Please add a short explanation for each picture (a, b, c, and d) of Fig. 14 in caption and
picture.
Reply:Modified as required
